# Estimation of Snow Depth over the Qinghai-Tibetan Plateau Based on AMSR-E and MODIS Data

**Liyun Dai** [1,3,4]**, Tao Che** [1,2,*]**, Hongjie Xie** [3] 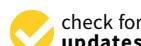 **and Xuejiao Wu** [5]

[1] Key Laboratory of Remote Sensing of Gansu Province, Heihe Remote Sensing Experimental Research Station, Cold and Arid Regions Environmental and Engineering Research Institute, Chinese Academy of Sciences, Lanzhou 730000, China; dailiyun@lzb.ac.cn

[2] Center for Excellence in Tibetan Plateau Earth Sciences, Chinese Academy of Sciences, Beijing 100101, China

[3] Laboratory for Remote Sensing and Geoinformatics, Department of Geological Sciences, University of Texas at San Antonio, San Antonio, TX 78249, USA; hongjie.xie@utsa.edu

[4] Jiangsu Center of Collaborative Innovation in Geographical Information Resource Development and Application, Nanjing 21003, China

[5] State Key Laboratory of Cryospheric Sciences, Cold and Arid Regions Environmental and Engineering Research Institute, Chinese Academy of Sciences, Lanzhou 730000, China; xjwu@lzb.ac.cn

* Correspondence: chetao@lzb.ac.cn; Tel.: +86-0931-4967366

**Abstract:** Snow cover over the Qinghai-Tibetan Plateau (QTP) plays an important role in climate, hydrological, and ecological systems. Currently, passive microwave remote sensing is the most efficient way to monitor snow depth on global and regional scales; however, it presents a serious overestimation of snow cover over the QTP and has difficulty describing patchy snow cover over the QTP because of its coarse spatial resolution. In this study, a new spatial dynamic method is developed by introducing ground emissivity and assimilating the snow cover fraction (SCF) and land surface temperature (LST) of the Moderate Resolution Imaging Spectroradiometer (MODIS) to derive snow depth at an enhanced spatial resolution. In this method, the Advanced Microwave Scanning Radiometer-Earth Observing System (AMSR-E) brightness temperature and MODIS LST are used to calculate ground emissivity. Additionally, the microwave emission model of layered snowpacks (MEMLS) is applied to simulate brightness temperature with varying ground emissivities to determine the key coefficients in the snow depth retrieval algorithm. The results show that the frozen ground emissivity presents large spatial heterogeneity over the QTP, which leads to the variation of coefficients in the snow depth retrieval algorithm. The overestimation of snow depth is rectified by introducing the ground emissivity factor at 18 and 36 GHz. Compared with in situ observations, the snow cover accuracy of the new method is 93.9%, which is better than the 60.2% accuracy of the existing method (old method) which does not consider ground emissivity. The bias and root-mean-square error (RMSE) of snow depth are 1.03 cm and 7.05 cm, respectively, for the new method; these values are much lower than the values of 6.02 cm and 9.75 cm, respectively, for the old method. However, the snow cover accuracy with depths between 1 and 3 cm is below 60%, and snow depths greater than 25 cm are underestimated in Himalayan mountainous areas. In the future, the snow cover identification algorithm should be improved to identify shallow snow cover over the QTP, and topography should be considered in the snow depth retrieval algorithm to improve snow depth accuracy in mountainous areas.

**Keywords:** snow depth; passive microwave; Qinghai-Tibetan Plateau; emissivity; land surface temperature; snow cover fraction; snow depletion curve

## 1. Introduction

Snow cover on the Qinghai-Tibetan Plateau (QTP) contributes a large portion of the water supply for the large rivers in Asia [1], reduces the incident solar radiation absorbed at the surface, inhibits heat stored in the ground from being released to the overlying atmosphere, and subsequently influences climate change [2], which will further influence ecosystems and spring phenology [3–5]. Reportedly, snow cover increases soil moisture, and has opposite effects to global warming on alpine plant phenology and reproductive success over the QTP [6]. Therefore, it is important to investigate the temporal and spatial variations of snow cover over the QTP for hydrological, climate, and ecosystem change studies.

Meteorological stations over the QTP operated by the China Meteorological Administration (CMA) have been providing time series of snow depth data since the 1950s. However, these stations are mainly distributed in the Eastern part of the QTP and in areas with low elevations. These inhomogeneous distributions limit the application of these data in climate and hydrological research. Satellite-based optical remote sensing has been successfully used to derive snow cover in clear sky conditions since 2000 [7–10], but it cannot monitor snow depth. Although there are some reanalysis and assimilation products containing a snow depth layer, the accuracy of these products depends on inputs (especially precipitation), and their spatial resolutions are too coarse (approximately $0.6°\sim1°$) [11–13].

Currently, passive microwave (PMW) remote sensing is widely used to derive snow depth. There are four PMW snow depth/snow water equivalent (SWE) products covering the QTP: NASA's Advanced Microwave Scanning Radiometer-Earth Observing System (AMSR-E), standard SWE product [14], the European Space Agency's (ESA) GlobSnow SWE product [15,16], the snow depth product from the West Data Center of China (WESTDC) [17], and the CMA's Fengyun snow depth product [18]. All of these methods were developed based on the original baseline retrieval algorithm (Chang method) by which the snow depth is assumed to be linearly dependent on the brightness temperature difference between 18 GHz and 36 GHz (TBD), and coefficients used in the method, which was developed based on the fundamental radiative transfer theory [19], changed with snow characteristics.

The AMSR-E product was produced from AMSR-E data using brightness temperature differences between 10 GHz and 18 GHz, and 10 GHz and 36 GHz for vertical polarization, to derive snow depth in the non-forest areas, and TBD to derive snow depth in forest areas. The polarization difference was adopted as the coefficient in this algorithm to solve the uncertainties caused by changes in snow characteristics. The SWE was produced by multiplying snow depth by snow density [20]. This product overestimated snow cover over the QTP [21]. GlobSnow was produced by ESA from SMMR, SSM/I and SSMI/S data. This method assimilated in situ snow depth to optimize snow grain size using the Helsinki University of Technology (HUT) model [22]. The dynamic relationship between snow depth and simulated TBD was built based on the optimized snow particle size. The SWE was also calculated by multiplying snow depth and snow density [22]. Due to the assimilation of in situ snow depth, the accuracy of GlobSnow is higher than that of the AMSR-E product, but GlobSnow did not cover the QTP. The WESTDC snow depth was calculated based on the modified Chang algorithm, which is an empirical equation expressed as the relationship between in situ snow depth and TBD, and monthly offsets were used to decrease the influence caused by the change in snow characteristics [17]. The Fengyun snow depth was calculated by CMA from Microwave Radiation Imagery (MWRI) data, also an empirical local method. The relationships between in situ snow depth and brightness temperature differences were established over different land types [18]. Because the two local algorithms utilized the snow depth from meteorological stations, their accuracies were higher than those of the AMSR-E product.

These products adopted different strategies to remove the influence of variations in snow characteristics but did not consider the influence of ground emissions. They used the same snow cover identification method (Grody's decision tree) [23], and this identification method presented overestimation of snow cover over the QTP [21,24]. In the Grody decision tree, TBD was used to

discriminate scatterers and non-scatterers. When TBD is greater than 0 K, scatterers are believed to exist. The thresholds of the TBD and polarization differences were set to discriminate other scatterers (cold desert and frozen ground) from snow cover. The simple threshold method cannot discriminate the snow-free areas that produce large TBD from snow-covered areas, and may sometimes omit snow cover because the TBD produced by snow-free areas varies with ground emissivity [25]. Ground emissivity depends on soil characteristics, varying temporally and spatially, and a large (small) difference in ground emissivity at 18 and 36 GHz leads to a large (small) TBD. If the areas with a large TBD from the ground are covered by snowpack, the TBD collected by the sensor is influenced by both ground and snow cover. In this case, the existing method did not separate these two contributions, leading to overestimation of snow depth.

Furthermore, the QTP is characterized by patchy snow, and the PMW snow depth with a grid size of approximately 25 km cannot precisely describe the distribution of snow depth. This coarse spatial resolution also makes validation difficult. To improve the spatial resolution of snow depth, a method was developed to downscale the PMW snow depth based on PMW snow depth and MODIS snow cover products, and this method worked well in Alaska, USA [26]. However, the accuracy of this method depends on the accuracy of snow depth and snow cover products. Due to the low accuracy of PMW snow depth caused by ground emissivity over the QTP, the application of this method may also result in large errors. Therefore, it is necessary to remove the errors caused by ground emissivity to improve the snow depth accuracy in the QTP.

In this study, a spatial dynamic method combining snow cover fraction (SCF), ground emissivity, and PMW brightness temperature is developed to derive snow depth with an enhanced spatial resolution over the QTP. The data used in this study are introduced in Section 2. In Section 3, a detailed description of the snow depth retrieval methodology is described, and the results and validation are clarified in Section 4. In Section 5, we discuss the uncertainties of this method, and in the final section, a simple conclusion of this work is provided.

## 2. Data

### 2.1. Passive Microwave Brightness Temperature

The AMSR-E instrument on the Aqua satellite from the National Aeronautics and Space Administration (NASA) Earth Observing System (EOS) provides global PMW brightness temperature data. AMSR-E was launched on 2 May 2002 and ceased operations on 4 October 2011. The frequencies included 6.9 GHz, 10.7 GHz, 18.7 GHz, 23.8 GHz, 36.5 GHz, and 89.0 GHz for both horizontal and vertical polarizations (Table 1). In this study, brightness temperature data at 18.7 GHz, 23.8 GHz, 36.5 GHz, and 89 GHz from 2003 to 2010 are used to retrieve snow depth over the QTP.

The Advanced Microwave Scanning Radiometer-2 (AMSR2) onboard the Global Change Observation Mission (GCOM-W) has provided brightness temperature data since 3 July 2012. AMSR2 was the continuation of AMSR-E and has the same channels as AMSR-E but a slightly smaller footprint (Table 1). The AMSR2 brightness temperatures are used to derive the snow depth for the field work period in March 2014.

The MWRI on FY-3B has provided brightness temperature at 10.65 GHz, 18.7 GHz, 23.8 GHz, 36.5 GHz, and 89 GHz for both horizontal and vertical polarizations since 5 November 2010. These footprints of MWRI are coarser than those of AMSR-E (Table 1). In this study, the MWRI brightness temperatures are used to supplement the blank AMSR2 data in March 2014. To avoid errors caused by snow melting, only night overpass time brightness temperatures are used.

**Table 1.** Footprints for the frequencies of AMSR-E on EOS-Aqua, MWRI on FY3 and AMSR2 on GCOM-W.

| Sensor | AMSR-E | AMSR2 | MWRI |
|---|---|---|---|
| Satellite | EOS-Aqua | GCOM-W | FY3B/C |
| Frequency: Footprint (GHz): (km × km) | 10.65: 29 × 51<br>18.7: 16 × 27<br>23.8: 18 × 32<br>36.5: 8.8 × 14.4<br>89:4 × 4.5 | 10.65: 24 × 42<br>18.7: 14 × 22<br>23.8: 15 × 26<br>36.5: 7 × 12<br>89: 3 × 5 | 10.65: 51 × 85<br>18.7: 30 × 50<br>23.8: 27 × 45<br>36.5: 18 × 30<br>89: 9 × 15 |

*2.2. MODIS SCF*

The Collection-5 Terra/Aqua MODIS Level 3, 500 m daily fractional snow cover products (MOD10A1 and MYD10A1) were archived in the National Snow and Ice Data Center (NSIDC) [7]. This dataset is stored in HDF and contain 10 layers of data, among which the SCF [27,28] and cloud mask [29] layers are used in this study. The SCFs are key data in the spatial dynamic algorithm of snow depth retrieval and are used to identify snow cover areas. Both SCF and cloud mask data are used to identify snow-free PMW grids, which is a vital step in calculating ground emissivity, an important variable in the snow depth retrieval algorithm.

On the QTP, the average cloud coverage was approximately 47.3%, which limited the usefulness of MODIS snow cover products. Therefore, in this study, the no-cloud snow cover fraction from 2003 to 2011 was obtained by using a temporal cubic spline interpolation method to remove cloud cover [30], which was used to identify snow cover over the QTP.

*2.3. MODIS Land Surface Temperature (LST)*

The Collection-6 MODIS LST product (MOD11A1) with a spatial resolution of 1 km was generated by the generalized split-window LST algorithm [31]. In this study, MOD11A1 is used to calculate the ground emissivity over the QTP at different microwave frequencies. To maintain temporal consistency with the night overpass orbit of AMSR-E, only night-time LST data are used in this study.

*2.4. Snow Depth from Meteorological Stations*

The CMA operates 123 meteorological stations over the QTP. Snow depth and snow density are regularly observed at every station. The positions of these stations are shown in Figure 1 (green dots). Snow depth is measured at local 8:00 am every day, and snow density is measured every five days. In this study, daily snow depths from 2003 to 2010 are used to validate the estimated snow cover and snow depth. Based on the snow density observations, the average snow density is approximately 150 kg/m$^3$ over the QTP.

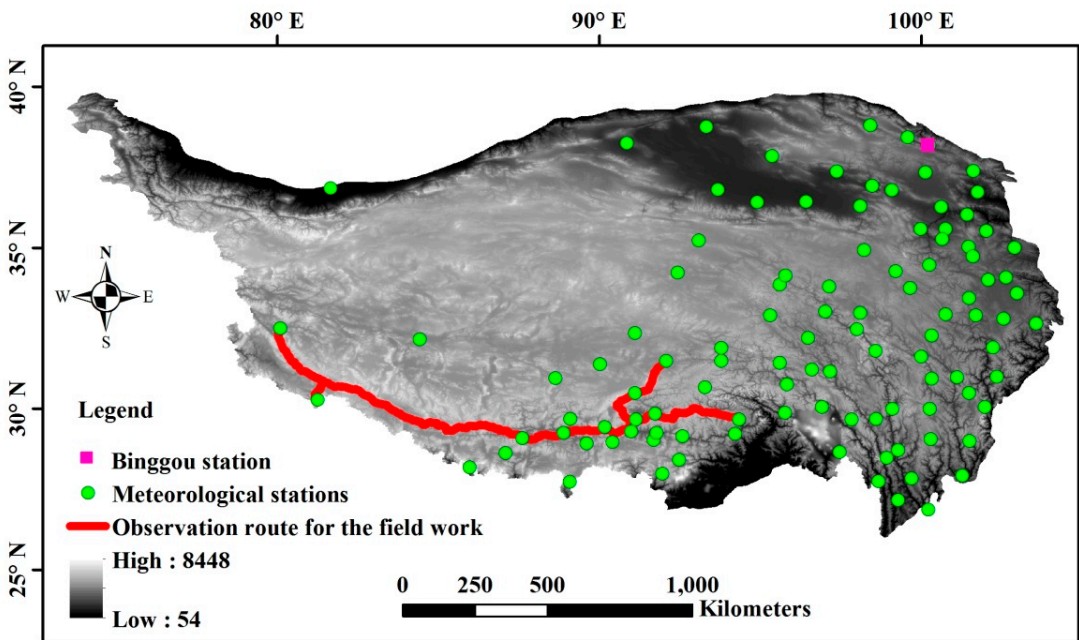

**Figure 1.** Distribution of meteorological stations of the China Meteorological Administration (CMA) (green dots) over the Qinghai-Tibetan Plateau (QTP), the snow observation route for field work in March 2014 (red line), and the location of Binggou station (pink square). The background shows elevation.

### 2.5. Snow Depth Observed during Field Work

Snow depths were measured along an observation route with a length of approximately 1600 km (Figure 1, red line) from 23 March to 31 March 2014. They were measured every 5~10 km in the snow-cover area, and a total of 56 snow depths were obtained. Among these snow depths, ten of them were 0 cm, and the average snow depth was 7.1 cm. These observational data are used here to validate the estimated snow depth.

Snow depth and soil temperature at 0 cm and 4 cm depths since 2013 were automatically measured at the Binggou station in the Qilian mountainous area (Figure 1, pink square). Snow depth and soil temperature were measured by an ultrasonic sensor (SR50A) and temperature sensor (Campbell 109S), respectively. In this study, these data were used in the discussion section to clarify the uncertainty caused by the ground emissivity calculations.

### 3. Methodology

In this study, snow depths with a spatial resolution of 500 m are retrieved using the AMSR-E brightness temperature, MODIS SCF product, and MODIS LST product. The process is summarized in Figure 2. The Grody decision tree is used to divide the AMSR-E grids into snow-free and snow-covered grids [23]. It is possible that snow cover is not detected by AMSR-E when the snow depth or SCF is small in an AMSR-E grid. Therefore, if the grid is identified as snow-free by AMSR-E, the MODIS SCF data are used again to identify snow cover, and the subgrid snow depth (sd_M) is calculated based on MODIS SCF (scf_M) using the snow depletion curve (SDC). If the grid was identified as a snow cover grid by AMSR-E, the AMSR-E brightness temperature and MODIS SCF are combined to derive sd_M based on the new spatial dynamic method. When developing the new method, the snow depth retrieval algorithm in a pure snow-covered grid is first developed and then used in the development of an algorithm for a mixture grid. The algorithm in a pure snow-covered grid will be described in Section 3.1, and that in a mixture grid is described in Section 3.2. The SDC will be introduced in Section 3.3.

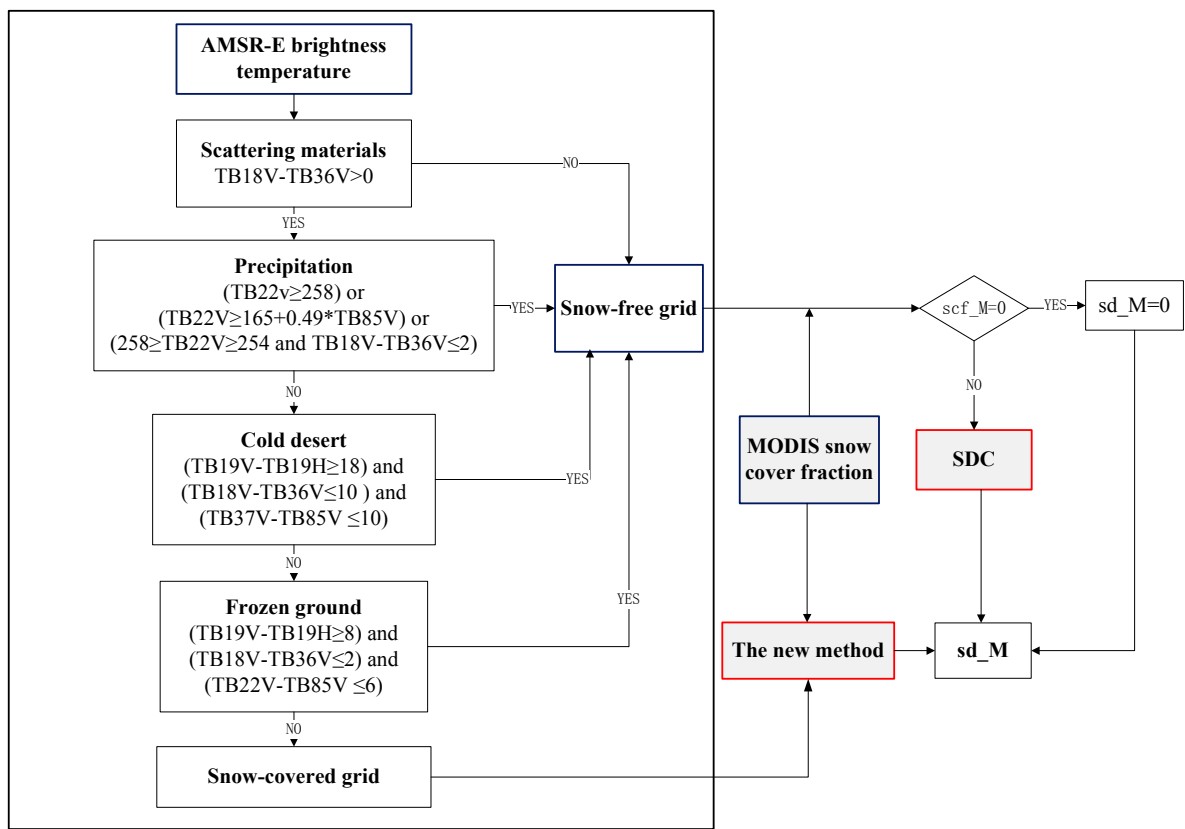

**Figure 2.** Flowchart of snow depth (sd_M) derivation with enhanced spatial resolution over the QTP. The Grody decision is described in the dashed box [23].

### 3.1. Snow Depth Retrieval in a Pure Snow-Covered Grid

The brightness temperature gradient method is a simple and efficient way to derive snow depth from PMW data [19,32] and is described as follows:

$$\text{SD} = a \cdot (\text{TB}_{18,s} - \text{TB}_{36,s}) - b, \tag{1}$$

where SD is estimated snow depth and the coefficients *a* and *b* vary with snow characteristics and ground emissivity at 18 GHz and 36 GHz. The subscript "s" indicates the snow-covered area. $\text{TB}_{18,s}$ and $\text{TB}_{36,s}$ are the brightness temperatures of snow-covered areas at 18 GHz and 36 GHz, respectively. Because the vertical polarization is more sensitive to scatterers [23], both frequencies for vertical polarization are used in Equation (1) in this study. The coefficient "*a*" varies with snow characteristics, especially snow grain size. Although snow characteristics are important factors in the snow depth derivation and a change in these characteristics will cause a change in coefficient "*a*", the focus of this study is on the ground emissivity, and the snow characteristics are set as average values.

The existing methods usually set coefficient "*b*" as 0 K [19,22,32,33], which means that the ground emissions at 18 GHz and 36 GHz are the same. However, the ground emission may change with frequency because of the variation in ground emissivity with frequency, and a coefficient *b* of 0 K led to a serious overestimation of snow cover over the QTP [21,25]. Therefore, in this study, the microwave emission model of layered snowpacks (MEMLS) [34] is used to simulate the brightness temperature at different ground emissivities, obtaining the coefficients "*a*" and "*b*" that change with ground emissivity. MEMLS was developed based on radiative transfer, using six-flux theory to describe multiple volume scattering and absorption. It can simulate the brightness temperature of layered snowpack with different snow characteristics (thickness, temperature, grain size, snow density, and liquid water content in each layer).

### 3.1.1. Influence of Ground Temperature on the Snow Depth Derivation

MEMLS is used to simulate the brightness temperature at 18 GHz ($TB_{18,s}$) and 36 GHz ($TB_{36,s}$) for varying snow depth, emissivity, and snow-covered ground temperature of the top layer ($T_{g,s}$), and is used to obtain the varying coefficients in Equation (1), which describes the relationship between snow depth and ($TB_{18,s} - TB_{36,s}$). Table 2 describes the ranges of variable parameters. Among the parameters, $T_{g,s}$ changes from 260 to 270 K at an interval of 1 K. Ground emissivity at 18 GHz ($\varepsilon_{18}$) changes from 0.9 to 1.0 at an interval of 0.01. The soil/snow interface reflectivity at 18 GHz (re18) equals (1-$\varepsilon_{18}$). The reflectivity difference between 18 and 36 GHz (re18–re36) changes from 0 to 0.005 at an interval of 0.001. Snow depth changes from 1 to 30 cm at an interval of 1 cm. The other snow parameters used in the MEMLS are set as constants (Table 3). The snowpack is set as one layer, the correlation length is 0.1 mm, both liquid water content and salinity are 0, snow density is set as 150 kg/m³, and snow temperature is 265 K.

**Table 2.** Ranges and steps of variables used for the simulation of brightness temperature by MEMLS.

| Items | $T_{g,s}$ | $\varepsilon_{18}$ | re18 | re36-re18 | Snow Depth |
|---|---|---|---|---|---|
| Range | 260~270 K | 0.9~1.0 | 0.0~1.0 | 0~0.05 | 1~30 cm |
| Interval | 1 K | 0.01 | 0.01 | 0.001 | 1 cm |

**Table 3.** The constant values of snow parameters used for the simulation of brightness temperature by MEMLS.

| Items | Correlation Length | Liquid Water Content | Snow Density | Snow Temperature |
|---|---|---|---|---|
| Value | 0.1 mm | 0 | 150 kg/m³ | 265 K |

The coefficients "*a*" and "*b*" in Equation (1) vary with $T_{g,s}$, re18, and re36. Based on the simulation results, $T_{g,s}$ has a slight influence on the relationship between snow depth and ($TB_{18,s} - TB_{36,s}$) (Figure 3).

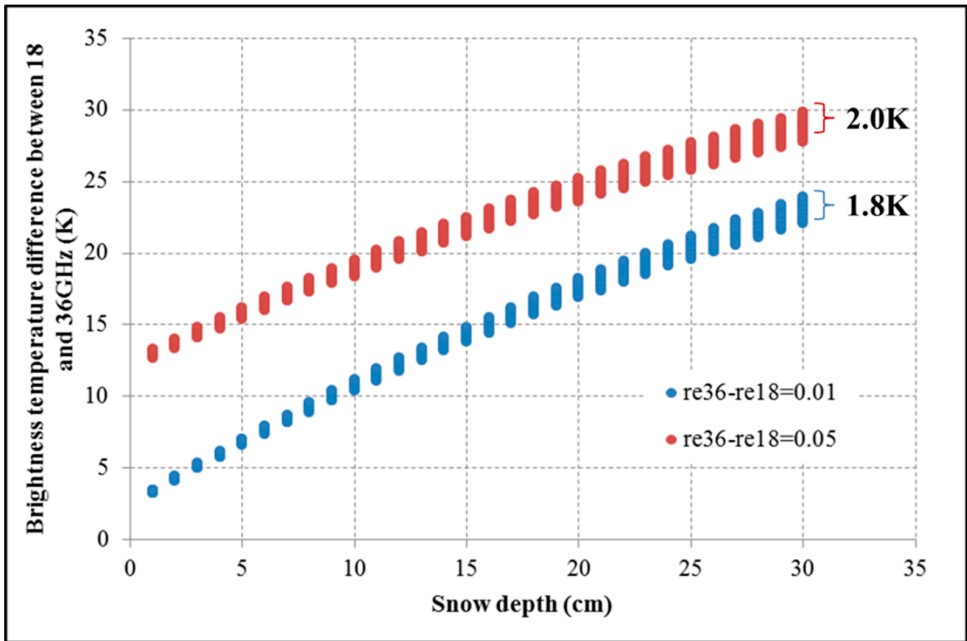

**Figure 3.** The brightness temperature difference between 18 and 36 GHz ($TB_{18,s} - TB_{36,s}$) changes with snow depth and $T_{g,s}$, given that the soil/snow reflectivity at 18 GHz (re18) is 0.03, the correlation length is 0.1 mm, the snow density is 150 kg/m³, and the reflectivity differences (re36-re18) are 0.01 and 0.05, respectively.

Given that re18 equals 0.03, $TB_{18,s}$ and $TB_{36,s}$ are simulated with changes in snow depth and $T_{g,s}$ when re36-re18 equals 0.001 and 0.005, respectively (Figure 3). Figure 3 shows that $(TB_{18,s} - TB_{36,s})$ varies with $T_{g,s}$ over a very small range for a fixed snow depth. When the snow depth is 30 cm, the difference of $(TB_{18,s} - TB_{36,s})$ from $T_{g,s}$ = 270 K and $T_{g,s}$ = 260 K arrives at the maximum with 1.8 K and 2.0 K if re18-re36 equals 0.01 and 0.05, respectively, and this difference declines with decreasing snow depth. The influence of $T_{g,s}$ on the relationship between snow depth and $(TB_{18,s} - TB_{36,s})$ is so small that it is ignored in this study.

### 3.1.2. Coefficients Vary with Ground Emissivity

Since the influence of $T_{g,s}$ on the snow depth derivation can be ignored, the coefficients "*a*" and "*b*" in Equation (1) are obtained when $T_{g,s}$ is set as 265 K. Their variation with re18 and re36-re18 are exhibited in Figure 4. The coefficient "*a*" increases with re18 and re36-re18. The coefficient "*b*" increases with re36-re18, but it is slightly influenced by re18. Therefore, if the $\varepsilon_{18}$ and $\varepsilon_{36}$ are determined, the coefficients "*a*" and "*b*" can be obtained, and then the snow depth retrieval algorithm (Equation (1)) is determined.

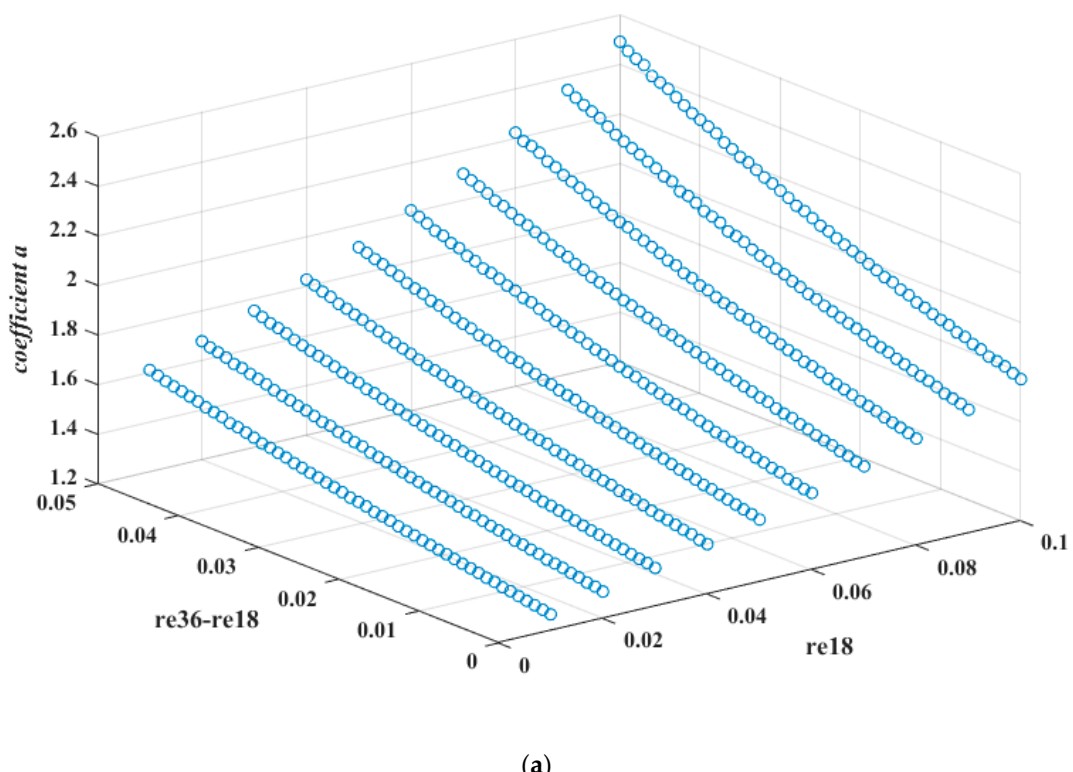

(**a**)

**Figure 4.** *Cont.*

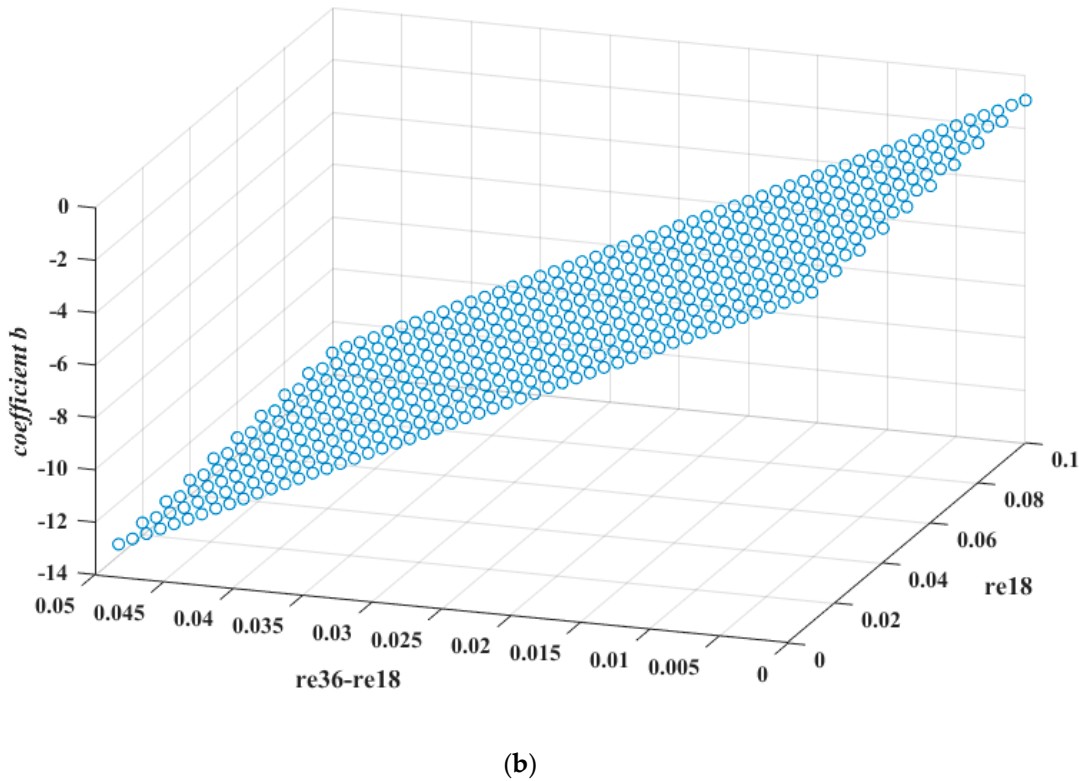

**(b)**

**Figure 4.** Coefficients "*a*" and "*b*" vary with re 18 and re18-re36 when $T_{g,s}$ is set as 265 K. (**a**) indicates coefficient "*a*"; (**b**) indicates coefficient "*b*".

### 3.1.3. Determination of Ground Emissivity and Coefficients

In this study, $\varepsilon_{18}$ and $\varepsilon_{36}$ are calculated using MODIS LST and AMSR-E brightness temperatures at 18 and 36 GHz. First, the MODIS LST with a spatial resolution of 1 km (LST_M) is aggregated to that with a spatial resolution of 25 km (LST_A) by averaging the LSTs in an AMSR-E grid because the spatial resolution of the brightness temperature and LST are different. Then, MODIS SCF data are used to discriminate snow-covered, snow-free, and cloud-covered grids. Only the data of the snow-free AMSR-E grid with LST_A less than 270 K are used to calculate emissivity, in order to avoid the errors brought by thaw soil. In this study, the average emissivities of each AMSR-E grid from 2003 to 2008 are used in the snow depth retrieval process, and the expressions of these emissivities are described as follows:

$$\varepsilon_f = \frac{1}{nd} \sum_{d=1}^{d=nd} \frac{TB_{f,d}}{LST\_A_d}, \text{ if } LST\_A_d < 270K \text{ and } scf\_A_d = 0, \tag{2}$$

where $\varepsilon_f$ is the ground emissivity at frequency f, $TB_{f,d}$ is the brightness temperature on the dth day at frequency f, and LST_A$_d$ is the LST on the dth day in an AMSR-E grid. "nd" is the total number of days meeting the condition that LST is less than 270 K, and SCF is 0, from 2003 to 2008. Figure 5 presents the spatial distribution of $\varepsilon_{18}$ and $\Delta\varepsilon$.

Lower $\varepsilon_{18}$ and larger $\Delta\varepsilon$ values are found in the Western and Northern areas compared with the other areas of the QTP (Figure 5). Large $\Delta\varepsilon$ means large TBD contributed by snow-free ground, which leads to the overestimation of snow cover from the existing method. Based on our previous study, the Northwest area exhibits the largest commission errors over the QTP, which are larger than 60% [25], where the $\Delta\varepsilon$ presents the largest value of all over the QTP (Figure 5). There are some pixels with no data meeting the condition "LST < 270 K and SCF = 0", most of which contain lakes or glaciers. In the Southeastern part of the QTP (25~28°N, 92~100°E) with low elevation, the LSTs are more than 270 K.

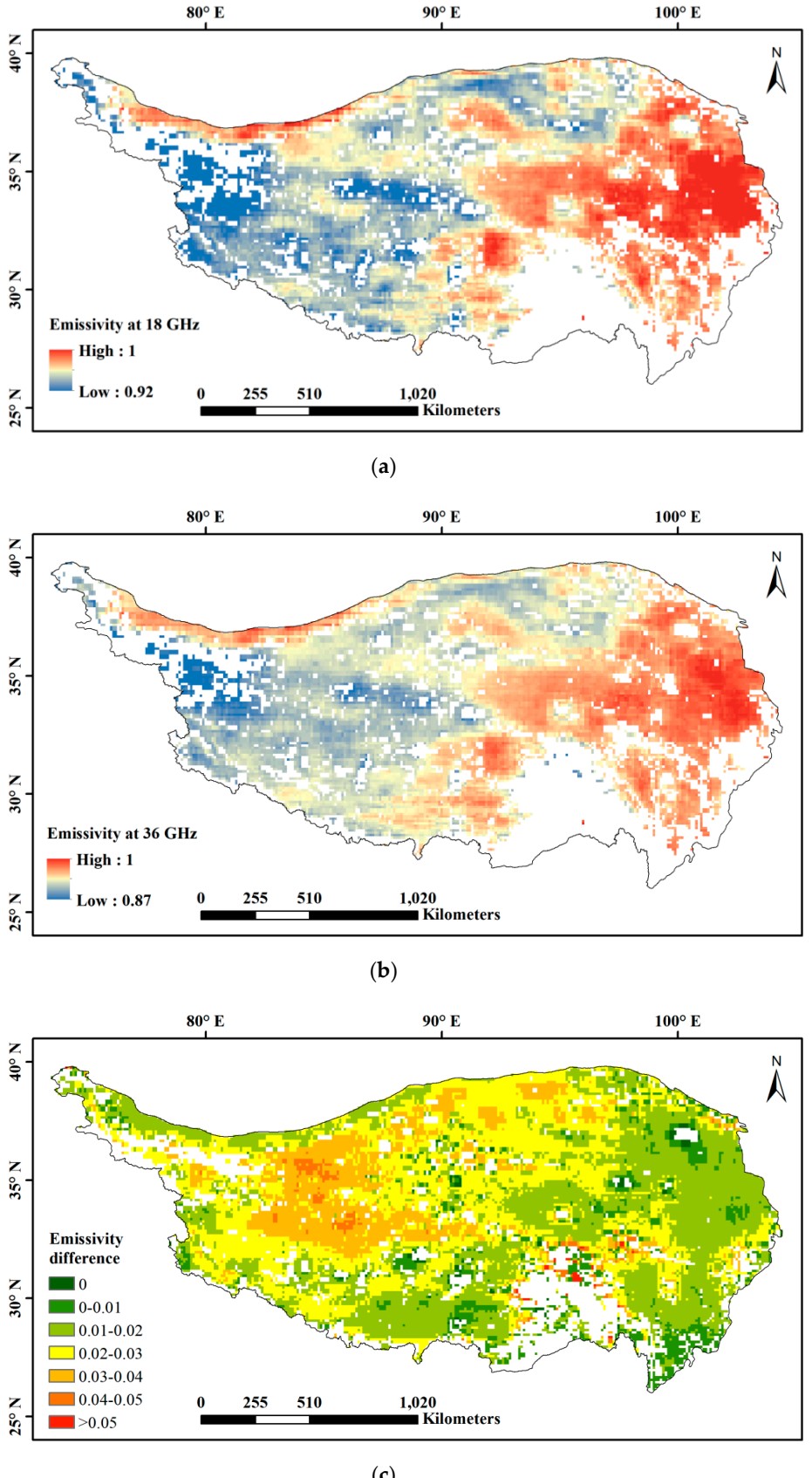

**Figure 5.** Distribution of the emissivity at 18 GHz ($\varepsilon_{18}$) (**a**) and 36 GHz ($\varepsilon_{36}$) for vertical polarization (**b**), and emissivity difference between 18 and 36 GHz ($\Delta\varepsilon$) for vertical polarization (**c**) over the QTP.

Since $\varepsilon_{18}$ and $\varepsilon_{36}$ are determined, the coefficients "*a*" and "*b*" in Equation (1) can be obtained as in Figure 6. The blank grids in the figures are permanent snow cover, lake areas or an area with LSTs larger than 270 K. In the lake areas or high LST areas, PWM shows no snow. For the permanent snow cover areas, the $\varepsilon_{18}$ and $\varepsilon_{36}$ are assumed to be 1.0, based on which the coefficients "*a*" and "*b*" are 1.25 and 0, respectively.

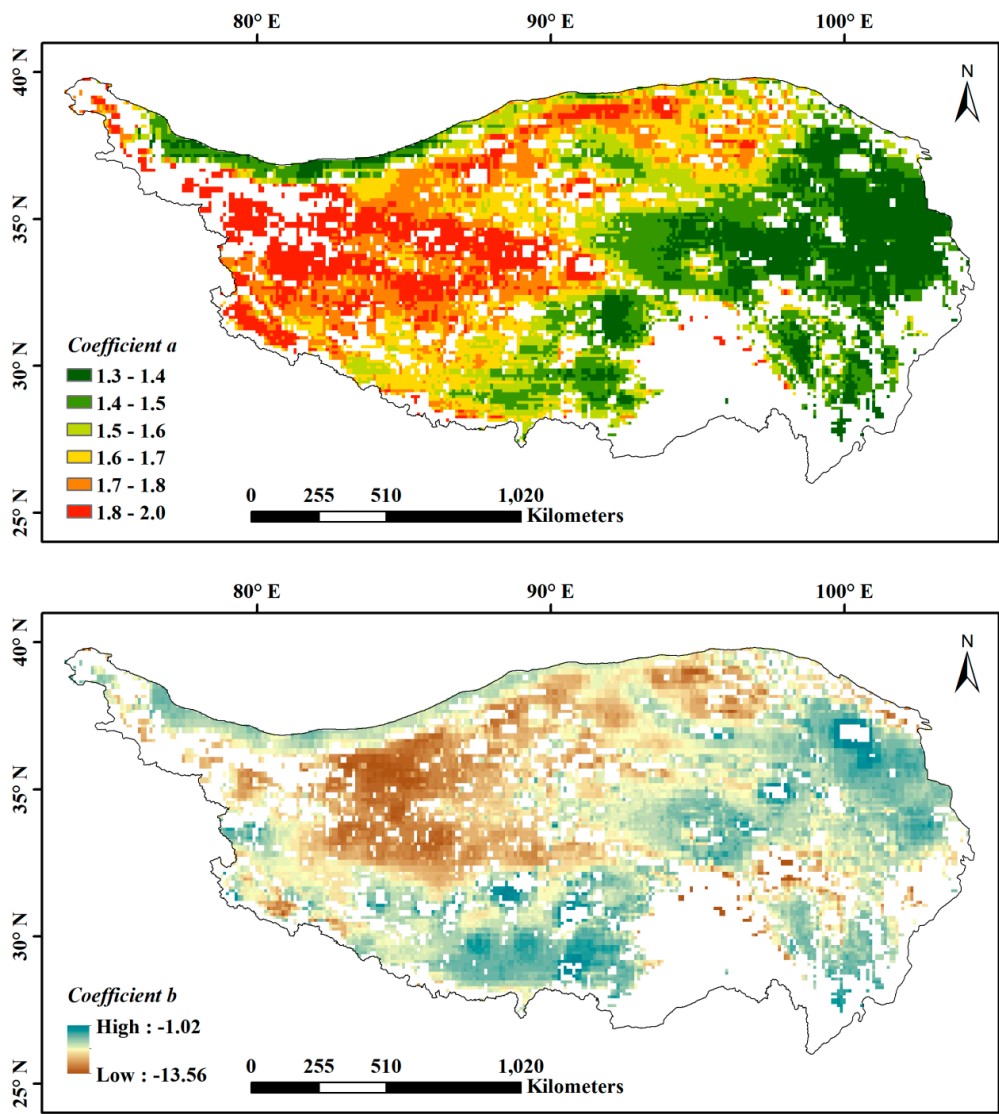

**Figure 6.** Distribution of coefficients "*a*" and "*b*" over the QTP.

The polarization difference (PD) of 18 GHz is also derived over the QTP, and the areas with PD larger than 18 K present low emissivity and large $\Delta\varepsilon$. From Figure 5, $\varepsilon_{18}$ is between 0.93 and 0.95, and $\Delta\varepsilon$ is between 0.3 and 0.4 in the area. Based on Figure 4, the coefficient "*b*" in Equation (1) should be between 8.6 K and 11.1 K. These values are consistent with the result of Grody's decision tree of snow cover identification, in which the area is identified as snow-free when the PD is larger than 18 K and the TBD is less than 10 K [23].

### 3.2. Snow Depth Retrieval in a Mixture Grid

For a mixture grid (AMSR-E pixel, 25 km) with snow-covered and snow-free subgrids (MODIS pixel, 500 m), the brightness temperature can be expressed as follows:

$$\mathrm{TB_f} = \mathrm{T_{g,n}} \cdot \varepsilon_f \cdot (1 - \mathrm{scf\_A}) + \mathrm{TB_{f,s}} \cdot \mathrm{scf\_A}, \tag{3}$$

where $T_{g,n}$ is the ground temperature without snow cover, and subscript "n" indicates snow-free areas. This value comes from the MODIS LST product at night time. $TB_{f,s}$ is the brightness temperature of the snow-covered area at frequency f, and subscript "s" indicates snow-covered areas. $\varepsilon_f$ is the ground emissivity at frequency f, and it was determined in Section 3.1. $TB_f$ is brightness temperature at f frequency in a mixture grid, and it is obtained from AMSR-E data. scf_A is SCF in an AMSR-E grid, and it is calculated through Equation (4).

$$scf\_A = \frac{\sum_{i=1}^{i=m} scf\_M_i}{m}, \tag{4}$$

where $scf\_M_i$ is the SCF of the $i^{th}$ subgrid in an AMSR-E grid. m indicates the number of subgrids in an AMSR-E grid.

Then, the TBD in a mixture grid ($TB_{18} - TB_{36}$) can be expressed as follows:

$$TB_{18} - TB_{36} = T_{g,n} \cdot \varepsilon_{18} \cdot (1 - scf\_A) + TB_{18,s} \cdot scf\_A) - (T_{g,n} \cdot \varepsilon_{36} \cdot (1 - scf\_A) + TB_{36,s} \cdot scf\_A) \tag{5}$$

Then, the ($TB_{18,s} - TB_{36,s}$) is described as follows:

$$TB_{18,s} - TB_{36,s} = \frac{TB_{18} - TB_{36} - (1 - scf\_A) \cdot T_{g,n} \cdot (\varepsilon_{18} - \varepsilon_{36})}{scf\_A}, \tag{6}$$

Based on Equation (1), the snow depth of the snow-covered area (SD) can be expressed as follows:

$$
\begin{aligned}
SD &= a \cdot \left( \frac{TB_{18} - TB_{36} - (1 - scf\_A) \cdot T_{g,n} \cdot (\varepsilon_{18} - \varepsilon_{36})}{scf\_A} - b \right) \\
&= a \cdot \left( \frac{TB_{18} - TB_{36} - T_{g,n} \cdot \Delta\varepsilon}{scf\_A} + T_{g,n} \cdot \Delta\varepsilon - b \right),
\end{aligned} \tag{7}
$$

where $\Delta\varepsilon = \varepsilon_{18} - \varepsilon_{36}$, the coefficients "*a*" and "*b*" depend on re18 and re18-re36 and have been determined in Section 3.1 (Figures 5c and 6).

After the snow depth of the snow-covered area is calculated, the snow depth of the snow-covered part of the subgrid was SD, and the remaining part was snow-free, namely, snow depth = 0. Therefore, the snow depth of the whole subgrid (sd_M) can be calculated based on the SCF of the subgrid (scf_M) and expressed as Equation (8).

$$sd\_M = SD \cdot scf\_M, \tag{8}$$

### 3.3. SDC (Snow Depletion Curve)

Because of the coarse spatial resolution, it is possible that AMSR-E cannot detect the snow cover of the grid with small SCF, and in this case, the snow depth information cannot be obtained based on AMSR-E. To address this, SDC is used to derive snow depth based on SCF data. SDC is defined as the relationship between snow depth and SCF, and it is used to calculate the snow depth of subgrids where AMSR-E shows snow-free characteristics, but MODIS SCF is more than 0. There are over ten SDCs that were developed in different regions [35]. Among these SDCs, the SDC developed by Niu and Yang is dynamic and a function of snow density [36]. This SDC is adopted in this study to estimate snow depth, and the snow density is set as 150 kg/m$^3$, which is the average snow density over the QTP (Figure 7).

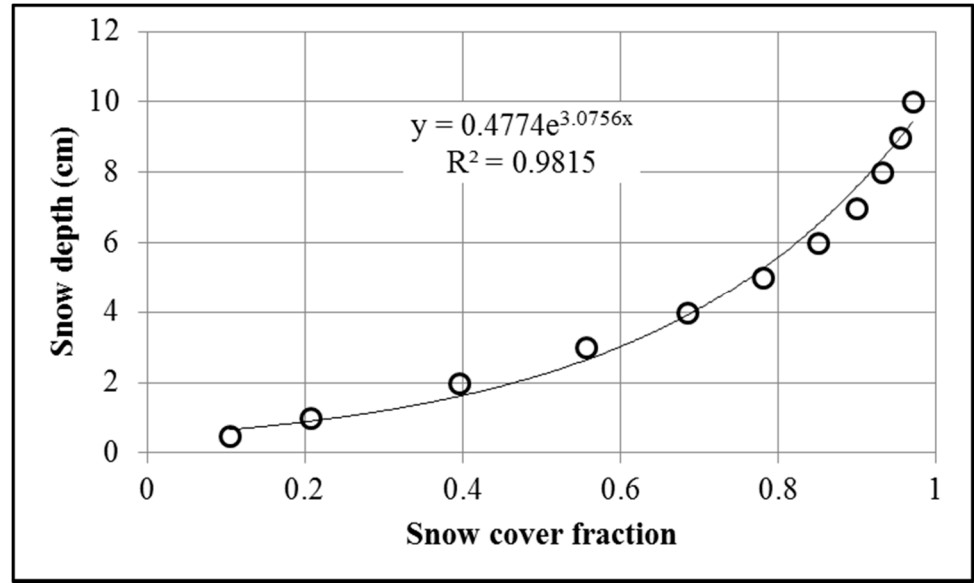

**Figure 7.** Variation in snow depth with snow cover fraction (SCF) snow depletion curve (SDC), given a snow density of 150 kg/m³.

## 4. Results and Validations

### 4.1. Results

Both the SDC and dynamic algorithm were used to derive daily snow depth from AMSR-E from 2003 to 2010, and the seasonal average snow depth was calculated from the daily snow depth. Figure 8 presents the spatial distribution of snow depth in spring (March, April and May; MAM), summer (June, July and August; JJA), autumn (September, October and November; SON) and winter (December, January and February; DJF). Most areas are characterized by ephemeral snow over the QTP. Deep snow is mainly distributed in the Himalaya, Pamir, and Southeastern mountainous areas.

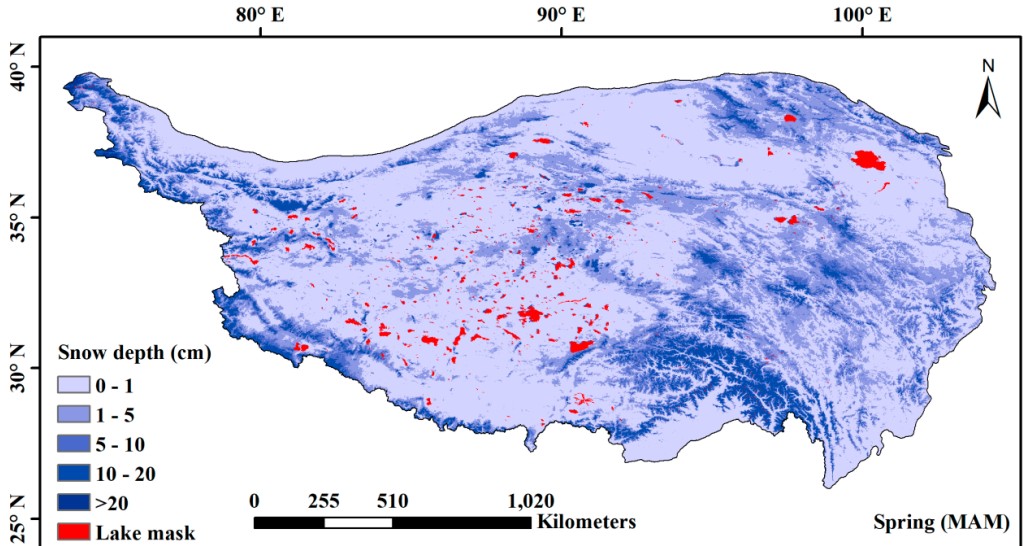

**Figure 8.** *Cont.*

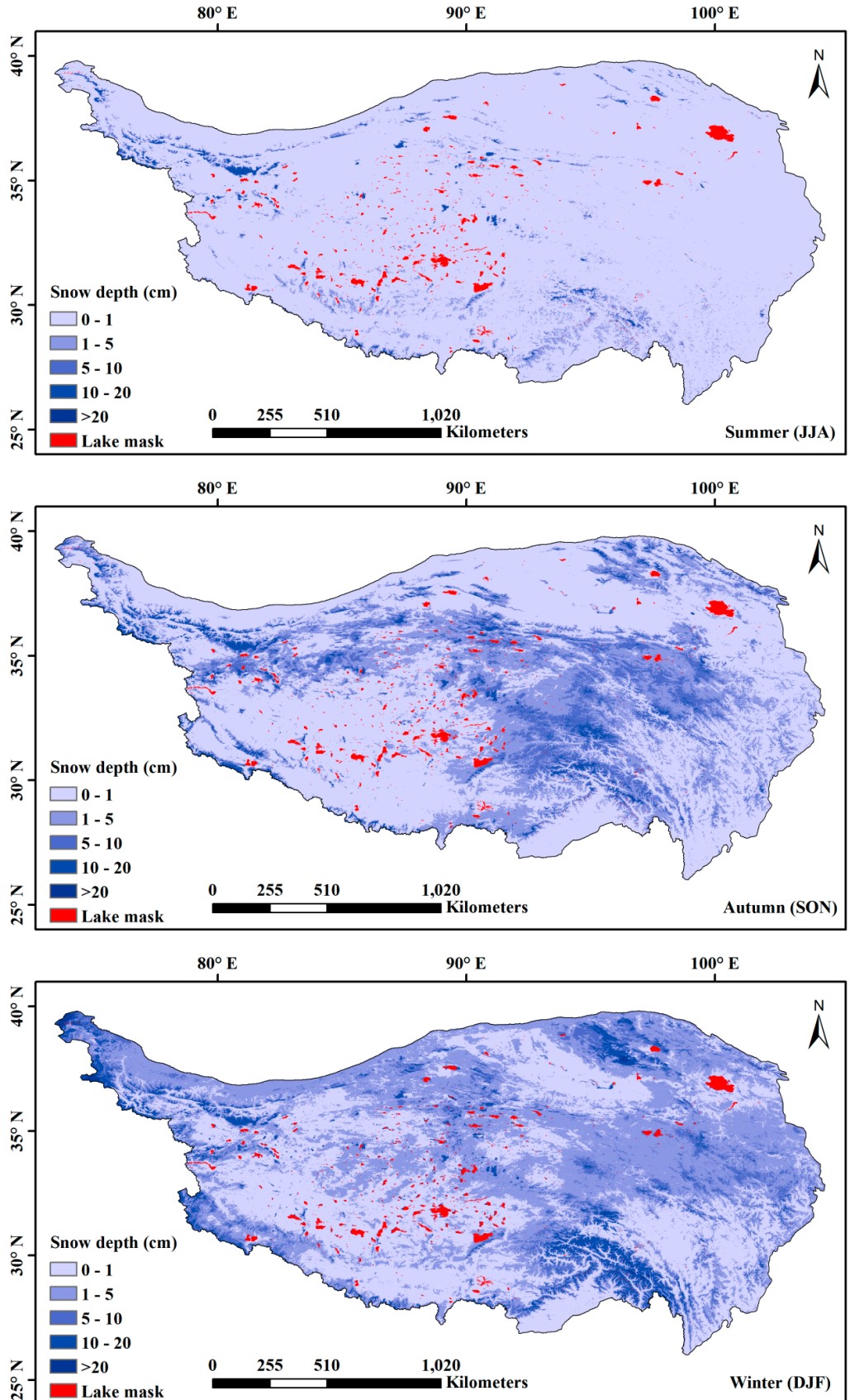

**Figure 8.** Spatial distribution of the average snow depth derived from AMSR-E and Moderate Resolution Imaging Spectroradiometer (MODIS) products using the dynamic algorithm for four seasons from 2003 to 2010 and lake distribution over the QTP.

## 4.2. Validation

The snow depths observed at meteorological stations of the CMA from 2003 to 2010 and during the field work in March 2014 are compared with the derived snow depth using the new spatial dynamic method (new method) and the modified Chang algorithm developed by Che [17], which did not consider the ground emissivity (old method).

### 4.2.1. Identification of Snow Cover

First, the snow depths observed at meteorological stations are used to validate the accuracy of the estimated snow cover. The results show that the overall accuracy of snow cover is 93.9%. The snow cover accuracy is close to 1 when the snow depth is 0 cm and less than 70% when the snow depth is ≤2 cm. When snow depth is >2 cm, the accuracy increases with the snow depth (Figure 9a).

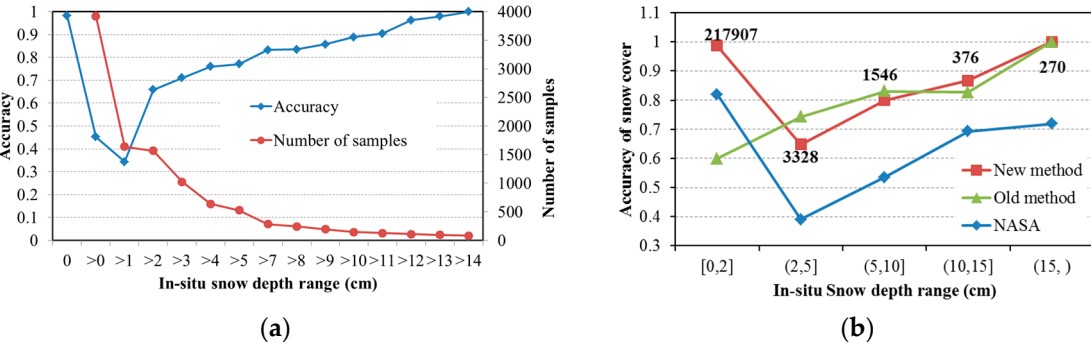

(**a**)                           (**b**)

**Figure 9.** (**a**) Variation in the snow cover accuracy and sample numbers with the in situ snow depth from 2003 to 2010 (the number of samples for 0 cm snow depth is 89,624), and (**b**) comparison of snow cover accuracy between the old and new methods.

The accuracy of snow cover from the new method, old methods, and the NASA product are compared, and the results show that the new method can efficiently identify snow-free areas with an accuracy of 98.4%, but the old method and NASA product seriously overestimated snow cover with an accuracy of 75.0% and 82.1% over the QTP. When the snow depth is between 2 and 5 cm, the new method shows larger omission errors with an accuracy of 64.8%, compared with the old method with an accuracy of 74.2%, but lower than the NASA product of 39.0%. When the snow depth is larger than 5 cm, the old and new methods show close accuracies, both over 80%, but the NASA product still exhibits a large omission error (Figure 9b). The overall identification accuracy of snow cover from the new method, old methods, and NASA product are 93.9%, 60.2%, and 80.1%, respectively.

### 4.2.2. Retrievals of Snow Depth

The accuracy of the NASA snow depth product, snow depth derived using the old method without considering the ground emissivity, and the new spatial dynamic method are compared based on meteorological station observations (Figure 10) and field work observations (Figure 11). Figure 10 demonstrates that the estimated average snow depth from the new method agrees well with the in situ snow depth when the snow depth is less than 25 cm. The new method shows much smaller bias and RMSE than those of the old method and NASA product when snow depth is less than 15 cm (Table 4). The accuracies of the new and old methods are close to each other when the snow depth is more than 15 cm, and their biases and RMSEs are less than that of NASA product. However, they present underestimation when snow depth is greater than 25 cm (Table 4). The overall bias and RMSE are 0.09 cm and 1.34 cm for the new method, 1.88 cm and 5.74 cm for the old method, and 0.61 cm and 5.12 cm for the NASA product, respectively. When compared with all in situ snow depths, they are 1.03 cm and 7.05 cm for the new method, 6.02 cm and 9.75 cm for the old method, and 2.24 cm and

9.62 cm for the NASA product, respectively, when compared with the in situ snow depth of more than 0 cm.

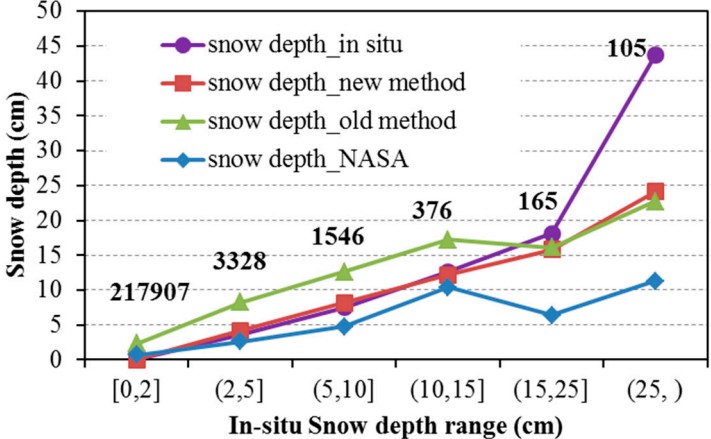

**Figure 10.** Comparison of average snow depth from in situ observation and estimated by the old method and the new method in 6 snow depth ranges, with the sample size labeled in back numbers above the polyline.

Snow depths estimated using the new and old methods at the observation points in the field work are compared with the observed value, and the result is depicted in Figure 11. The new method shows a better relationship with the in situ observation than the old method, and the bias and RMSE values (−1.3 cm and 4.6 cm) are much lower than those of the old method (4.2 cm and 7.9 cm) (Table 4).

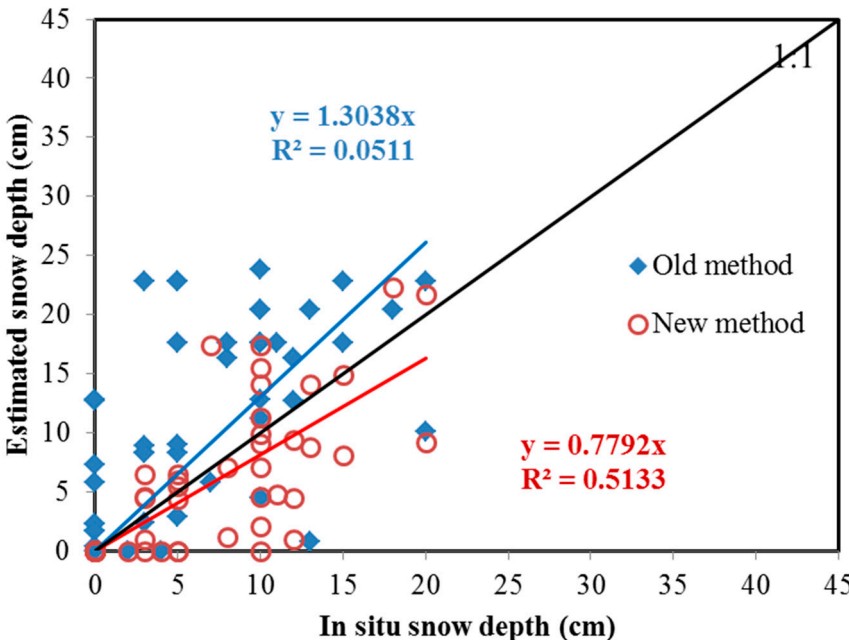

**Figure 11.** Comparisons between in situ snow depth and estimated snow depths from the new method and the old method at the observed points in the field work on March 2014.

**Table 4.** Error statistics from the implementation of the new and old methods, and NASA product to the field observation data from 23 to 30 March 2014 and meteorological station data.

| Data Range of In Situ Observation (cm) | Number of Samples | Observed Mean Snow Depth (cm) | New Method | | Old Method | | NASA | |
|---|---|---|---|---|---|---|---|---|
| | | | Bias (cm) | RMSE (cm) | Bias (cm) | RMSE (cm) | Bias (cm) | RMSE (cm) |
| Field work | 50 | 7.1 | −1.3 | 4.6 | 4.2 | 7.9 | 2.4 | 5.6 |
| [0, 2] | 217,907 | 0.0 | 0.0 | 1.1 | 2.3 | 4.1 | 0.7 | 2.7 |
| (2, 5] | 3328 | 3.7 | 0.6 | 6.3 | 4.6 | 9.6 | −1.0 | 5.4 |
| (5, 10] | 1546 | 7.5 | 0.7 | 6.6 | 5.1 | 8.9 | −2.7 | 8.8 |
| (10, 15] | 376 | 12.6 | −0.4 | 6.9 | 4.9 | 7.0 | −2.2 | 15.9 |
| (15, 25] | 165 | 18.1 | −2.1 | 8.1 | −1.3 | 8.12 | −11.7 | 16.0 |
| (25, 70] | 105 | 43.7 | −19.6 | 28.8 | −20.9 | 30.9 | −32.4 | 31.6 |
| All data | 223,427 | 0.12 | 0.09 | 1.34 | 1.88 | 5.74 | 0.61 | 5.12 |

The relative absolute bias (Rabias) and RMSE are calculated for each meteorological station from 2003 to 2010 using Equations (9) and (10), and the results are exhibited in Figure 12. When the observed snow depth ($sdo_i$) is 0 cm, the Rabias is set as 0 if the estimated snow depth ($sde_i$) is less than 1 cm and set as 1 if $sde_i$ is larger than 1 cm. Figure 13 shows that the Rabias and RMSE of the new method are generally lower than those of the old method. The average Rabias and RMSE of all stations are 8.9% and 1.1 cm for the new method and 47.4% and 6.0 cm for the old method. The minimum and maximum RMSEs are 0 cm and 10.6 cm for the new method and 0.2 cm and 22.6 cm for the old method. The minimum and maximum Rabias values are 0 and 163.1% for the new method and 0 and 276.4% for the old method.

$$\text{Rabias} = \frac{1}{n}\left(\sum_{i=1}^{n} |sde_i - sdo_i|/sdo_i\right), \tag{9}$$

$$\text{RMSE} = \sqrt{\frac{1}{n}\left(\sum_{i=1}^{n} (sde_i - sdo_i)^2\right)}, \tag{10}$$

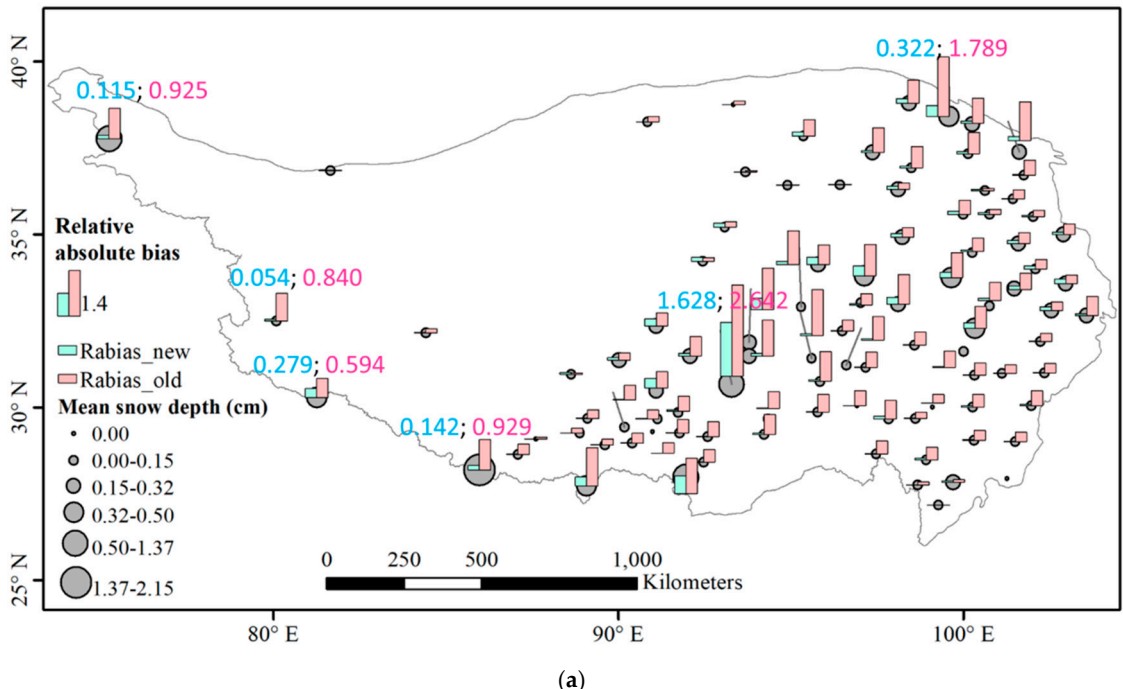

(**a**)

**Figure 12.** *Cont*.

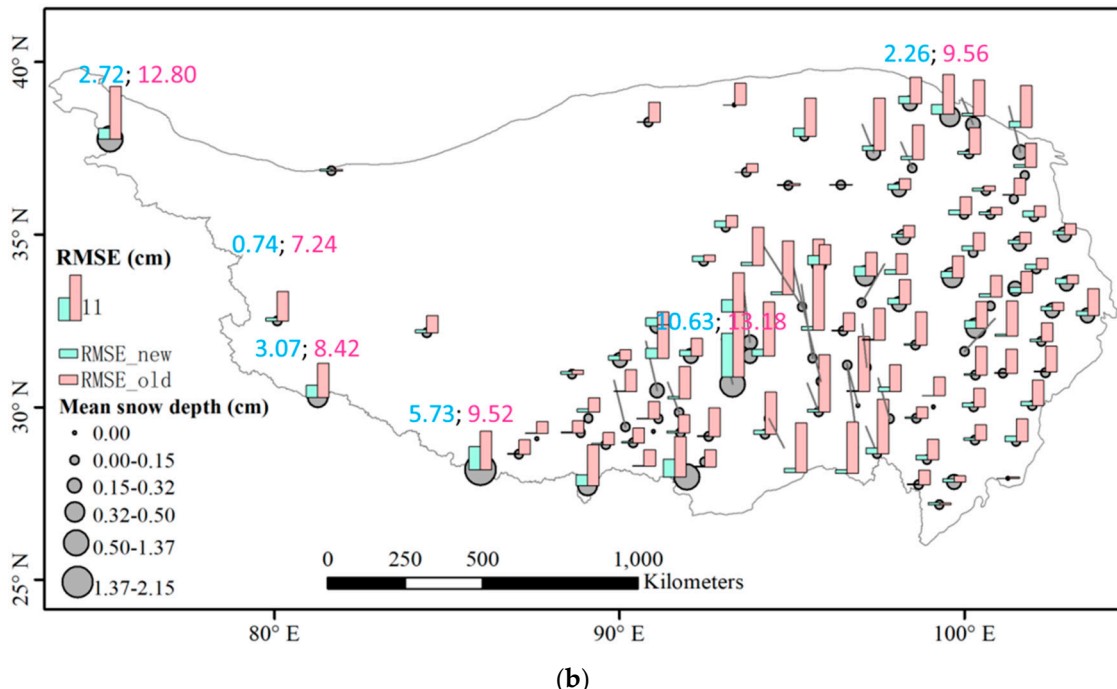

**Figure 12.** The mean observed snow depth is expressed as gray circles, with (**a**) relative absolute bias and (**b**) root-mean-square error (RMSE) for each station, with red bars for the old method and green bars for the new method, over the QTP from 2003 to 2010. The blue and red numbers label the values from the new and old methods, respectively.

Although the accuracy of calculated snow depth improved substantially compared with the old method, due to the application of ground emissivity, the RMSE remained very large when the snow depth was greater than 0 cm and less than 10 cm. The RMSEs are 6.3 cm and 6.6 cm for the snow depth range (2–5], with an average snow depth of 3.7 cm, and a range (5–10] with an average snow depth of 7.5 cm, respectively (Table 4). The ground emissivity depends on soil characteristics that change temporally, and thus the temporal static ground emissivity may cause errors.

## 5. Discussion

In this study, three kinds of uncertainties are related to the accuracy of derived snow depth, which are discussed in this section. The first is the calculation of ground emissivity, the second is the low accuracy of snow cover when snow depth is between 1 and 3 cm, and underestimation when snow depth is more than 25 cm. The third is the reason for the variation of ground emissivity with frequency.

### 5.1. Calculation of Ground Emissivity

In this study, the ground emissivity was calculated by dividing the brightness temperature by LST when the ground was not covered by snow and clouds, and the LST was less than 270 K. The ground emissivity should be determined by the effective soil temperature and brightness temperature. Here, the LST was used to replace the effective soil temperature because there is no available effective soil temperature.

The brightness temperature collected by the microwave radiometer comes from the upper layer soil. In the winter, the effective soil temperature of the upper layer is usually higher than the LST at night when the soil is frozen. Therefore, the actual emissivities should be less than the results obtained in this study. The in situ observation shows that the soil temperature difference between 0 cm and 4 cm was approximately 10 K when it was not covered by snow (Figure 13). If the difference between LST and effective soil temperature (the average temperature of the upper soil layer) is −5 K, the $\varepsilon_{18}$ will decrease by 0.017, and $\Delta\varepsilon$ will decrease by 0.0018 at most. The decrease of 0.0018 for the $\Delta\varepsilon$ will cause

decreases of approximately 0.5 K and 0.035 for the coefficients "*b*" and "*a*", respectively (Figure 4). The decrease of 0.017 for $\varepsilon_{18}$ will result in the increase of coefficient "*a*", which varies with $\varepsilon_{18}$ from 0.08–0.2 (Figure 4). This changing of coefficients will lead to biases of approximately 0.7 cm and 2.0 cm for snowpack with depths of 10 cm and 30 cm, respectively.

In addition, the ground under snow cover was assumed to be frozen ground, and the frozen ground emissivity was used for the retrieval of snow depth. Sometimes, the interface temperature between soil and snowpack is slightly higher than 0 cm, as on July 10 in Figure 14, and the ground emissivity may not be suitable. However, this usually occurs during the season with a relatively high temperature over the QTP, and the PMW cannot show the scattering characteristics of the snowpack. In the process of snow depth retrieval over the QTP, if AMSR-E does not detect snow in a grid, but MODIS finds it covered by snow, the snow depth is derived using SDC, which will not be influenced by the ground emissivity.

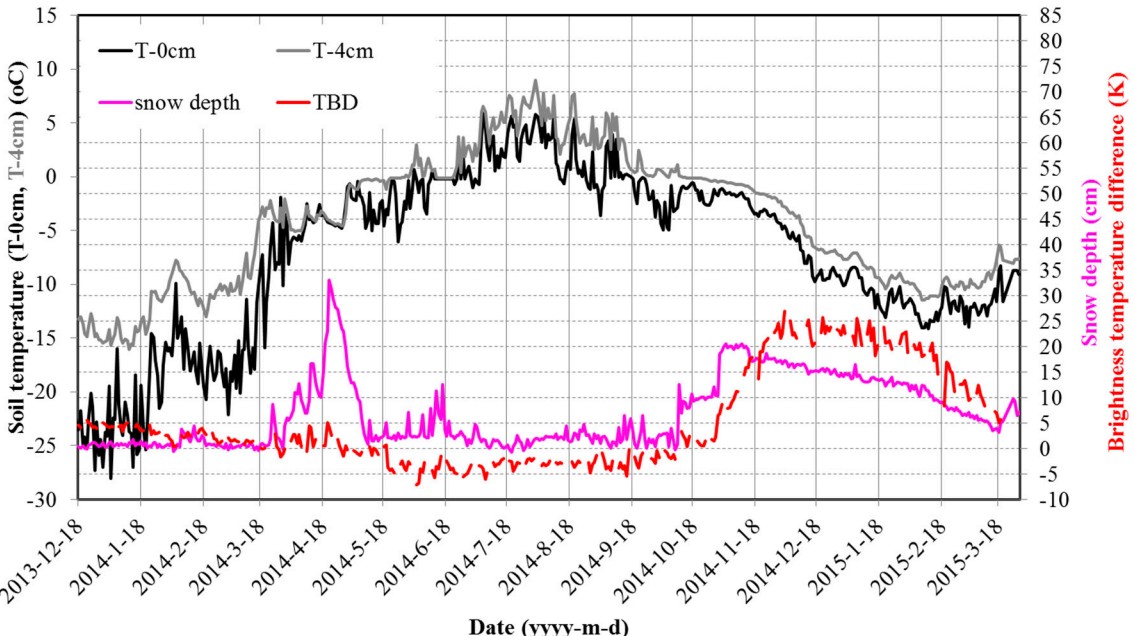

**Figure 13.** Temporal variations of soil temperature at 0 cm (T-0 cm) and 4 cm (T-4 cm), and snow depth and brightness temperature difference between 18 GHz and 36 GHz (TBD) at Binggou station from 18 December 2013 to 18 March 2015. The soil temperature and snow depth were measured by a temperature sensor (Campbell 109S) and ultrasonic sensor (SR50A), respectively. The brightness temperature difference was derived from AMSR-E observations. This figure shows that the soil temperature difference between 0 cm and 4 cm is larger for snow-free ground than for snow-covered ground.

### 5.2. The Omission of Shallow Snow and Underestimation of Deep Snow

Figures 9 and 10a show that snow depth between 0 and 3 cm presents the lowest accuracy of snow cover. There may be two reasons for the low accuracy of shallow snow depth. (1) It is difficult for MODIS to detect shallow snow cover over the QTP. The existing evaluation of MODIS snow cover over the QTP showed that the MODIS snow cover product is highly accurate when mapping snow with depth ≥4 cm, but has a very low accuracy for thin snow with depth <4 cm [37]. (2) The time gap between the in situ observation and MODIS scanning leads to a temporal mismatch. Due to the strong solar radiation, snow cover over the QTP melts quickly. The meteorological station observation time is 8:00 pm, and the overpass time of MODIS is approximately 10:00 am or 13:30 pm. Snow cover with snow depth less than 4 cm likely melts in 2 h (Figure 14a). Moreover, when deriving the long-term series of snow depths, the cloud-covered land was recovered using the temporal interpolation method.

The temporal interpolation possibly omits the cloud-masked ephemeral snow cover, as shown in Figure 14b.

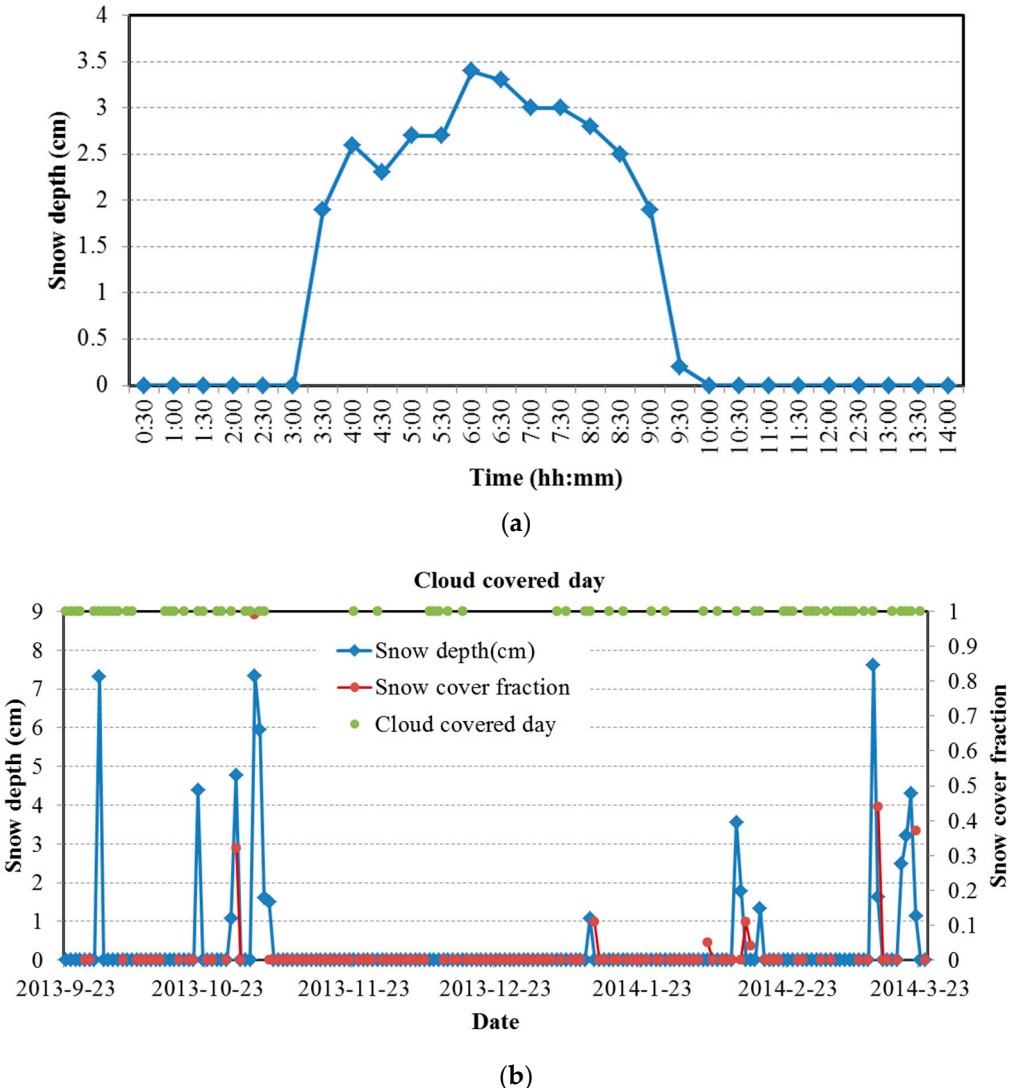

**Figure 14.** (**a**) Diurnal variation of snow depth at Cuona station (longitude: 93.23°E; latitude: 27.994°N) on 30 March 2014. (**b**) Daily variation of snow depth automatically measured by SR50A, MODIS SCF, and cloud cover by day at Jiali station (longitude: 93.283°E; latitude: 30.667°N), from 23 September 2013 to 23 March 2014.

This method underestimated the snow cover with depths greater than 25 cm, which mainly occurs in the Himalaya mountainous area. In this area, the TBD is generally less than 10 K even when the snow depth is greater than 40 cm, and the relationship between snow depth and TBD is poor. The low accuracy may be caused by signal saturation [33,38] and complex topography [20,39–41].

When snow depth reaches a threshold, the TBD will not increase with snow depth. In this study, the threshold is approximately 50 cm. In the Himalaya mountainous area, the observed maximum snow depth reached up to 70 cm during the 2003–2010 period, which is beyond the threshold. Some studies used lower frequencies to avoid the saturation problem caused by deep snowpack [42–44], but the complex topography in mountainous areas could change the directions of microwave signals emitted from ground, and the signals received by sensors included direct and indirect signals, from which it is difficult to separate snowpack information from other signals [39–41]. These factors can all bring uncertainty to the snow depth derivation in mountainous areas.

*5.3. Reasons for Ground Emissivity Variation with Frequency*

Ground emissivity is determined by the dielectric characteristics of the ground, which depend on temperature, moisture and soil texture [44–48]. In this study, ground emissivity was calculated when the LST was less than 270 K, thus, the value is the emissivity of frozen ground. When soil is frozen, the dielectric characteristics of the soil change due to a decrease in moisture; thus, the ground emissivity increases, and the emissivity difference between low and high frequencies presents a positive value [48].

The soil texture of the Western QTP is dominated by sandy loams in the upper 10 cm layer [49], and these areas exhibited large PDs and $\Delta\varepsilon$, which are characteristics of the Gobi desert [46,47,50]. In the Gobi areas, the deeper soil temperature is higher than land surface temperature at midnight (cold overpass time of AMSR-E); and thus the brightness temperature at high frequencies with smaller penetration depths is lower than that at low frequencies with larger penetration depths. In addition, sand particles present weak scattering, which further increases the brightness temperature gradient [47].

Therefore, it may be inferred that the $\Delta\varepsilon$ in the QTP results from the scattering of sand particles and soil temperature gradient. Furthermore, with the increase of frozen depths, deeper temperatures can be detected by the sensor, and the microwave signal will experience a longer path, and thus these factors have more opportunities to influence the microwave signal. The distribution of frozen depths in the QTP has been shown in Reference [49], and it was found that the area with a large frozen depth also presents a large $\Delta\varepsilon$ distribution.

However, currently, it cannot be determined which factor contributes the dominant influence and how these factors work together in the emissivity difference. Therefore, in the future, ground-based microwave radiometer observations will be carried out to develop a suitable model for simulating the emissivity of the frozen ground.

## 6. Conclusions

The existing snow depth retrieval algorithm presented large uncertainties over the QTP, which is characterized by patchy snow cover. In this study, a spatial dynamic snow depth retrieval algorithm was developed to derive snow depth with an enhanced spatial resolution for the QTP. This algorithm introduced ground emissivity to improve the snow depth accuracy, which was calculated by dividing the AMSR-E brightness temperature by the MODIS LST. Snow depth that cannot be detected by AMSR-E is estimated by SDC to decrease omission errors. The results revealed that the coefficients in the snow depth retrieval algorithm changed with the ground emissivity. The areas with large $\Delta\varepsilon$ presented large coefficient "*b*" in the snow depth retrieval algorithm, and the discussion showed that large $\Delta\varepsilon$ may be caused by the combination of frozen soil scattering, the temperature gradient within the upper soil layer, and the frozen depth. Compared with the in situ snow depth, the snow cover accuracy of the new method is 93.9%, which is better than the 60.2% accuracy of the old method. The bias and RMSE calculations of the snow depth are 1.03 cm and 7.05 cm for the new method, which is much lower than the 6.02 cm and 9.75 cm measures from the old method which did not consider the ground emissivity.

Therefore, the ground emissivity difference between frequencies is the main factor causing the overestimation of snow cover by PMW on the QTP. This new method improves the retrieval accuracy and spatial resolution of snow depth on the QTP by introducing the ground emissivity measures, MODIS SCF and SDC.

However, there are some uncertainties in the spatial dynamic algorithm. The identification accuracy of shallow snow cover by MODIS is low, and the depth of deep snowpack is underestimated in mountainous areas because of the complex topography. Therefore, in future work, the local snow cover identification method from MODIS should be developed to improve the identification accuracy, and auxiliary data such as topography should be used to improve the snow depth derivation in

mountainous areas. To further clarify the $\Delta\varepsilon$, ground-based microwave radiometer observations should be carried out to develop a suitable emissivity model for simulating the temporal dynamic emissivity.

**Author Contributions:** Conceptualization, L.D. and T.C.; Methodology, L.D.; Software, L.D.;Validation, T.C; Formal Analysis, L.D.; Investigation, L.D.; Resources, L.D; Data Curation, L.D and T.C.; Writing-Original Draft Preparation, L.D; Writing-Review & Editing, C.T., H.X. and X.W.; Supervision, T.C.; Project Administration, C.T.; Funding Acquisition, C.T."

**Funding:** This research was funded by the National Natural Science Foundation of China (grant nos. 91547210, 41771389, and 41671351), CAS 'Light of West China' Program and the National Foundational Scientific and Technological Work Programs of the Ministry of Science and Technology of China (grant nos. 2017FY100502).

**Acknowledgments:** The author Y. Dai wants to thank the China Scholarship Council for funding her visiting study (2016–2017) at the University of Texas at San Antonio.

**Conflicts of Interest:** The authors declare no conflict of interest.

## Abbreviation

| Abbreviated Names | Interpretation |
| --- | --- |
| TBD | Brightness temperature difference between 18 GHz and 36 GHz |
| $T_{g,s}$ | Ground temperature covered by snowpack |
| $T_{g,n}$ | Ground temperature without snow cover |
| $\varepsilon_{18}$ | Ground emissivity at 18 GHz |
| $\varepsilon_{36}$ | Ground emissivity at 36 GHz |
| $\Delta\varepsilon$ | Ground emissivity difference at 18 and 36 GHz, equal to $\varepsilon_{18} - \varepsilon_{36}$ |
| re18 | Soil/snow interface reflectivity at 18 GHz, equal to $1 - \varepsilon_{18}$ |
| re36 | Soil/snow interface reflectivity at 36 GHz, equal to $1 - \varepsilon_{36}$ |
| re36-re18 | Reflectivity difference between 36 GHz and 18 GHz, equal to $\varepsilon_{18} - \varepsilon_{36}$ |
| $TB_{18}$ | Brightness temperature at 18 GHz over mixture grid (AMSR-E grid) |
| $TB_{36}$ | Brightness temperature at 36 GHz over mixture grid (AMSR-E grid) |
| $TB_{18,s}$ | Brightness temperature at 18 GHz over pure snow-covered grid |
| $TB_{36,s}$ | Brightness temperature at 36 GHz over pure snow-covered grid |
| $TB_{f,d}$ | Brightness temperature on the dth day at frequency f |
| scf_A | SCF in an AMSR-E grid |
| scf_M | SCF in a MODIS grid |
| LST_A | LST over an AMSR-E grid |
| LST_M | LST over a MODIS grid |
| $LST\_A_d$ | LST on the dth day over an AMSR-E grid |
| sd_M | Snow depth over a MODIS grid (subgrid) |
| SD | Snow depth over a snow-covered area |
| SCF | snow cover fraction |
| MODIS | Moderate Resolution Imaging Spectroradiometer |
| LST | land surface temperature |
| AMSR-E | Advanced Microwave Scanning Radiometer-Earth Observing System |
| AMSR2 | Advanced Microwave Scanning Radiometer-2 |
| MEMLS | microwave emission model of layered snowpacks |
| QTP | Qinghai-Tibetan Plateau |
| PMW | passive microwave |
| WESTDC | West Data Center of China |
| HUT | Helsinki University of Technology model |
| MWRI | Microwave Radiation Imagery |
| NSIDC | National Snow and Ice Data Center |
| EOS | Earth Observing System |
| NASA | National Aeronautics and Space Administration |
| GCOM-W | Global Change Observation Mission |
| SDC | Snow depletion curve |

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
