# Peer review of "Estimation of Snow Depth over the Qinghai-Tibetan Plateau Based on AMSR-E and MODIS Data"

_remotesensing, doi:10.3390/rs10121989_

Round 1
Reviewer 1 Report
General comments
This study evaluates a new method for estimation of snow depth at the spatial resolution of MODIS snow cover product (i.e. 500m). The methodology is based on assimilation of MODIS observations into microwave AMSR product and it is tested over the Qinghai-Tibetan Plateau and validated by using daily snow depth observations at 123 meteorological stations. The results indicate that the new method is improves the mapping for snow free areas, but is less accurate for patchy snow cover conditions.
Overall the study is interesting, and within the scope of the journal. However the clarity of presentation can be improved. I have numerous specific comments (please see below), which are related to this. I would suggest to describe more clearly what is presented as the main result and perhaps to extend the evaluation by showing more detailed assessment of the accuracies (in the form of seasonal evaluations – showing also some typical time series). I like the grouping of results according to the snow depth classes, but it will be interesting to add some more assessment indicating the variability in the mapping accuracy in terms of some other climate or physiographic attributes (e.g. elevation of stations, land use, mean annual/seasonal precipitation, aridity, etc), which can increase the transferability of results to other regions.
Specific comments
Text in Data section - l.122-134 is likely not related to the results of manuscript. Please remove.
Some basic information about the MEMLS model is missing but will be useful. The section 3.1.1 is very specific and for readers not familiar with the approach thus very difficult to read and understand.
L.265 – why 265K?
Sections 3.1.3, 3.2, 3.3.What is the effect of clouds in the MODIS datasets for estimation of the average emissivity and snow depth mapping?
L.358-366 reads like a repetition of the aims or methodology. Perhaps consider to remove and to present directly the results.
Figure 8. Does this figure show the results for the new method?
Results: snow cover accuracy assessment. What is the accuracy of MODIS snow cover product compared to in situ measurements at the meteorological stations? Why is the assessment in Fig.9 just for the period 2003-2004?
Results: snow depth validation: The evaluation in Fig.10 is not clear. Please consider to explain what the lines represent? (The figure caption is not very helpful here). The same applies for Figure 11. The number of points here seems to be very low. I was expecting to see the results from 123 stations in the period 2003-2010. It will be very interesting to see a comparison between observed and estimated snow depths for some representative stations in the form of time series. Will it be possible to relate the mapping accuracy at individual stations with some physiographic attributes (the stations represent)? E.g. elevation, land cover in the neighbourhood, etc. This may bring some more general information which can significantly increase the transferability of results to other regions. Will it be also possible to generalize the accuracy along the transect from 2014? (e.g. by using some additional information about the elevations/;landuse along the transect?)
Author Response
Thanks for your comments and suggestions. According to your suggestions, we mainly made the following modifications:
1 Some basic information about the MEMLS model was added in section 3.1.
2 The influence of cloud on the snow depth retrieval was explained.
3 CaptionS of Figure 10 and Figure 11 was modified to make them clearer, and the comparison between accuracy of NASA snow depth product and the new method were added in section 4.
4 The accuracy analysis of snow depth was performed based on elevation.
Please see the details point-by-point in the attchment.

Reviewer 2 Report
Review of “Estimation of snow depth over the Qinghai-Tibetan Plateau based on AMSR-E and MODIS data” by L. Dai et al.
Summary
This paper presents an enhanced method of retrieving snow cover information from passive microwave and visible imagery. It uses the brightness temperature and surface temperature over cold snow-free land to derive the land surface emissivity. This is then used to correct standard snow retrievals that do not account for surface emissivity. It also uses the higher spatial resolution MODIS to obtain estimates fractional snow-covered areas. The method is validated via comparison with in situ snow observations. The new method outperforms the old method, particularly in areas with thin snow cover where surface emissivity has a larger effect.
General Comment
This is quite a nice paper. The technique is interesting and clearly is an improvement by accounting for land surface emissivity. The methods are well-described and the results are clear. The validation studies are thorough and convincing. The only issues are some fairly minor edits for improved grammar, typos, and the need for some clarification. In particular, some places seem a bit hastily written and need a good edit to clean up some of the text. I recommend acceptance after minor revisions to address the comments below.
Specific Comments
65-73: Note that there are JAXA products for AMSR2 too. Since this manuscript is presenting and validating a new method, using AMSR-E is fine, but when listing various PM snow products, should try to be comprehensive.
65-73: It seems like snow depth and SWE are used interchangeably here, but they are not. The PM sensors are really observing SWE. Obtaining snow depth requires some knowledge/assumptions about the density. This should be made clear here.
76: “TBD” needs to be explicitly defined. I was able to deduce that it stood for the “brightness temperature difference” mentioned in the line above, but that should add a “(TBD)” after that phrase.
81: What is the “HUT model”? At a minimum, need to provide a reference, but would be good to also briefly describe.
85: I know the Chang algorithm is well-known within the snow community, but a reference should still be provided here.
88: Need to spell out what MWRI stands for the first time it is used.
106-107: I always think it’s important to note that while the grid resolution is 25 km, the actual input resolution (sensor footprint) is often different, and in the case of SSMI and SSMIS coarser. So, the resolution issue is even more problematic than the gridded resolution indicates.
115: Need to define “SCF” the first time it is used.
122-134: I think this got left in by accident – it looks like template guidance for authors. Replace with relevant introductory text for the section.
138: AMSR-E ceased (normal) operations on October 4, 2011, not December 4.
141: Curious why you used 2003 instead of 2002? The data start in June 2002, which seems like that would be sufficient to get data from 2002.
157, 165: Why Collection-5 for MOD10 products and Collection-6 for MOD11. I assume that was the current availability? It strikes me as odd to use different versions, but I understand that the processing changes are not simultaneous. I don’t see where they would significantly affect your results, but I think a note making clear that the products are from different collections and why would be good to include.
178-181: Please add in the length of the transect and/or the spacing between observations. These are important to note I think.
289: Not sure what you mean by “commission errors” – is it supposed to be “emission errors”?
344: Write out “SDC” in the section header. This also defines it, which is not otherwise done in the text until the caption for Figure 7.
358-364: Most of this is not results but simply a restatement of the methods. I don’t think this is necessary since that was just described previously. The only thing new is the last sentence in the paragraph (364-366) that describes how you obtained the results. You could simply say something like “Implementing the methods described above, both the SDC and the dynamic algorithm were used to derive…”
406-407: The underestimation is most likely due to saturation. You note that in the conclusion, but I think you should state it here as you present the results.
418: Figure 11 has an interesting feature that I see: there are several in situ snow depths at 10 cm for the new method that correspond to different estimated snow depths. I can understand the spread in the estimates, but I’m curious why so many in situ records at 10 cm. Are the in situ measurements discretized (it looks like there are other values with similar behavior), maybe at 1 cm intervals? But why are these so evident for the new method, but not the old method? Aren’t the in situ snow depths the same for both estimation methods?
Minor Comments:
25: “from” instead of “to”
27: “brightness” instead of “rightness”
31: “show” instead of “showed” – use present tense here
51: “acts on climate change” is awkward. Maybe “influences climate change”
52: also use present tense here: “increases”, “has”
94: “resulted in” instead of “presented”
96: use “greater than 0 K” or simply “>0 K” instead of “more than 0 K”
433: typo in the equation with a hanging parentheses. It looks like it should be in the numerator?
441: “the accuracy improved” – I think maybe some words are missing. The sentence doesn’t make sense as it is written.
448: “should be discussed” is awkward. Use “are relevant”.
Author Response
Thanks for your comments and suggestions. According to your suggestions, we mainly made the following modifications:
1 Captions of Figure 10 and Figure 11 were modified to make them clearer, and comparison between the accuracy of NASA snow depth product and the new method was added in section 4.2.
2 Some abbreviations were explained or defined at their first time.
3 The length of the transect and the spacing between observations in the description of field work were added in section 2.5.
Please see the details point-by-point in attached file.

Reviewer 3 Report
The manuscript entitled “Estimation of snow depth over the Qinghai-Tibetan Plateau based on AMSR-E and MODIS data” presents an interesting paper about global snow depth using optical and passive microwave. The manuscript could be published in Remote Sensing but some minor changes are required.
- Please cited the reference Pérez-Cuevas et al. 2018 about Snow Cover Area of MODIS
https://www.mdpi.com/2073-4441/10/5/619
This manuscript present some technical approach which might be interested to be commented in the new version of the manuscript.
Under an operational mode, the authors must be commented about the seasonal variations of distributed coef (a and b)
Figure 12 has to be improved because of the spatial error distribution can not be evidence and the scale error has to be improved.
Figure 13 is difficult to understand as is, please improve it.
Figure 14 duplicates the snow depth parameter
Author Response
Thanks for your comments. According to your suggestions, we mainly made the following modifications:
1 Tomás et al. (2018) was cited in this paper.
2 Comments on the new algorithm was added in the conclusions.
3 Figure 12, 13 and 14 were modified to make them more understandable.
Please see the details point-by-point in attached file
